# Inequity in the Access to eHealth and Its Decomposition Case of Poland

**DOI:** 10.3390/ijerph19042340

**Published:** 2022-02-18

**Authors:** Justyna Rój

**Affiliations:** Department of Operational Research and Mathematical Economics, The Poznań University of Economics and Business, Al. Niepodległości 10, 61-875 Poznań, Poland; justyna.roj@ue.poznan.pl

**Keywords:** equity, eHealth, information and communication technology, Theil index

## Abstract

The aim of this research is to analyze the disparities in the distribution of information and communication technologies and skills across geographically determined population groups and to identify the source of the inequity. Literature showed that the nature of e-Health has the potential to resolve health inequalities. However, its successful implementation depends on such factors as the accessibility of required technologies to all people, the existence of technical infrastructure as well as people having the necessary information and communication skills. Employment of the Theil index allowed us to measure and decompose the national inequality into both: between and within macro-regions differences. Data was collected from Statistics Poland. The results showed the existence of inequity and its drivers. The novelty of this research results from application of the Theil index in the field of eHealth and identification of the barrier in access to e-Health, which can be a basis for improvement in government policy.

## 1. Introduction

The World Health Organization proclaims in its Constitution [1] that “…the highest attainable standard of health is a fundamental right of every human being.” Therefore, the existence of health inequalities is a major obstacle in achieving health justice [2]. Health inequity appears if systematic differences in health could be avoidable by taking the appropriate action [3]. It requires the monitoring of a number of factors such as: gender, socioeconomic and education status, geographic location, racial and ethnic differences, access to healthcare, health resources and the quality of health care, all of which affect the achievement of health equity. However, health, and thus health equity, is determined not only by social determinants but also by decision-making processes, implemented policies and social norms, structures which exist at all levels in society and therefore effective interventions are required to be taken in all sectors and areas of society [4,5].

Currently, we are witnessing rapidly developing digital health technologies, which have the potential for both positive and negative impacts on health equity. The engagement of information and communication technologies (ICTs) in health and health care has changed the way information is collected, distributed and utilized by both patients and healthcare providers. Therefore, ICTs have become an important tool in resolving health inequity [2,6]. However, if they are implemented without equity considerations, then they can create the risk of exacerbation of inequities or the appearance of new ones. If people do not have the skills or access to computers and networks then they also cannot use such technologies effectively and that would govern exposure to health-related risks among other discrimination in the digital health [7]. This risk is particularly crucial not only in low- and middle-income countries but also in rural and underserved areas within developed countries [8]. In this way, not everyone would benefit equally from such technologies, which could even reinforce existing health disparities [9]. Therefore, implementation of any ICT into healthcare requires a particular attention focused on equity impacts [10].

So far, researchers have investigated a range of different aspects of eHealth from the perspective of health equity either in the context of one country as Bidmon and Terlutter [11] and Craig and Rhee [12] or several countries, i.e., Andreassen et al. [13] or Ćwiklicki et al. [14].

In the case of Poland, previous studies—in the area of eHealth—focused primarily on the usage of medical Internet and e-health services by a particular group of people, such as the elderly [15,16], and more generally on the trends in the use of the Internet for health purposes [17,18], then on the access to particular forms of eHealth, such as electronic health record [19] or online health technologies and mobile devices [20]. Some studies focused on nurses’ opinions on the application of eHealth and their IT competence [21] or patients’ opinions on eHealth [22]. Mainly, these studies were based on the representative group of the population. There is also literature on the barriers and opportunities in development of publicly funded eHealth [23] or effectiveness of tele-medical care [24]. Thus, there are no such structured studies on equity and eHealth for Poland, thus this research will allow filling in the gaps in existing literature.

Thus, the aim of this study is to analyze the disparities in the distribution of information and communication technologies and skills across geographically determined population groups and to identify the source of the inequity.

For this purpose, the Theil index was engaged as it also allows identifying of drivers of disparities/inequities. Data was collected from the Statistics Poland database; thus, this study covers all populations of Poland in the ages from 16 to 74. Results showed that in Poland, due to geographic status, people do not have a fair opportunity to achieve their full digital health potential as some slight inequities were found and the existence of within-macro-regions differences are their main drivers. Results confirm that it is of high importance to improve both health policy and digitalization policy.

The paper is organized as follows. After the introduction (Section 1), the theoretical background of the research is presented in Section 2. Next, Section 3 is devoted to the description of the materials and methods. In Section 4, the results are presented, followed by the discussion in Section 5. Finally, the conclusion is presented in Section 6.

## 2. Theoretical Background

The World Health Organization (2005) describes eHealth as “the cost-effective and secure use of information and communication technologies (ICTs) in support of health and health-related fields, including health-care services, health surveillance, health literature and health education, knowledge and research”. Ehealth is a part of digital health, which expands the concept of eHealth and includes “digital consumers, with a wider range of smart-devices and connected equipment” as well as other uses of digital technologies for health, such as artificial intelligence, big data, Internet of Things and robotics [25]. Eysenbach (2001) also underlined that eHealth is not only: “technical development, but also a state-of-mind, a way of thinking, an attitude, and a commitment for networked, global thinking, to improve health care locally, regionally, and worldwide by using information and communication technology” [26].

Since the early 2000s, increasing use of ICTs in supporting of health services in both developed and developing countries can be observed, which is the result of the next fundamental technological transformation in human history, i.e., the digital age [27,28]. There is evidence, that digital transformation of health care has the potential to improve health outcomes by enhancing better diagnosis, data-based treatment decisions, clinical trials, digital therapeutics, self-management of care and person-centered care. It also allows one to create more evidence-based knowledge, competencies and skills for professionals to support health care [25,29]. This influences the present and future shape of health systems [30].

Many governments and healthcare organizations have noticed that eHealth provides a lot of hope for the most deprived as it ensures more individual and better health promotion and care [2]. Moreover, eHealth is perceived as one of the important elements in solving the problem of increased demand for health care by the ageing population [31]. Hence, eHealth is recognized as a key factor improving the well-being of people and stimulating economic growth [32]. It creates added value for people and all healthcare systems—above all, in the form of an increase in efficiency and improvement in quality and equity [33]. It also has the potential to reduce the cost of the functioning of the healthcare system [34]. Therefore, the WHO issued a global strategy for 2020–2025 on digital health, which aims to support international efforts “to develop the infrastructure for information and communication technologies for health… (and) to promote equitable, affordable and universal access to their benefits” along with promoting the development of national digital health strategies [25].

Thus, eHealth also plays a crucial role in achieving universal health coverage (UHC) by, for example, providing services to remote populations and underserved communities through tele-health and mHealth [35,36]. Introduction of electronic health records (EHRs) facilitates the provision of accurate and timely patient information which enhances the diagnosis and treatment of patients [25]. The World Health Organization treats UHC as a strategic priority. This goal is also a part of the United Nations 2030 Agenda for Sustainable Development, which was adopted by many countries in 2015 and which officially came into force on 1 January 2016 [37]. UHC represents a common belief that all people should have access to the health services they need (health promotion, prevention, treatment, rehabilitation, and palliative care) without risk of financial ruin or impoverishment. However, it does not mean free access to every possible health service for every person but rather it emphasizes the importance of access to health services and information as a basic human right [25].

Universal Health Coverage (UHC) includes the promise of equity, as every individual could receive affordable and good quality health services according to their need [38,39]. The importance of equity implies the fair opportunity for everyone to achieve their full health potential regardless of demographic, economic, social or geographic status [40,41,42,43]. The World Health Organization perceives equity as an intermediate objective of UHC [44]. However, equity is not a natural consequence of UHC policy implementation as some policies pursued in the name of UHC may even worsen inequalities [45,46,47]. Therefore, policymakers would have to balance decisions between increasing efficiency, population health or stimulating economic growth and improving equity [39,48].

As the nature of ICTs has changed the way information is collected, distributed and utilized by healthcare providers and patients, eHealth has become an important tool in resolving health inequities [2,6]. However, if people do not have the appropriate skills or even access to computers and networks then they would not be able to use digital health technologies effectively [49]. Thus, not all population groups will benefit equally from them and this could even reinforce existing health disparities [9]. Instead of benefiting the most from health information, they would become those “who are the least likely to benefit from advances in information technology, unless political measures ensure equitable access for all” [49].

Such existing disparities in access to e-Health, including telemedicine has been highlighted by the pandemic [50,51,52]. Rapid implementation of digital health innovations was one of the main results of the public health crisis posed by the coronavirus disease (COVID-19). It allowed people to maintain social distance and obtain health care at the same time. Thus, minimizing exposure to infection [31]. Many people in the world, for the first time in their lives, used tele-consultation from their family doctor, because they had to comply with social distancing requirements, especially, virtual health care or televideo-enabled interactions between patients and health care providers. Poverty, barriers to digital health literacy, lack of access to digital health and poor engagement with digital health are recognized as the main factors that could contribute to health inequities and poor health outcomes [10].

Therefore, it is of high importance to measure inequity and trace progress in this regard when implementing any eHealth solutions [53,54] to reduce digital health disparities.

Inequality in health has attracted increasing interest from not only the public but also academic community [55]. Studies on the equity in health care generally explore the equity impact of any changes based on disaggregated data by gender, race/ethnicity, geographical area, socio-economic status culture, education or other social advantages. However, it must also be underlined that the measurement of health inequalities remains challenging and is an evolving concept [56,57,58,59].

Therefore, this research is devoted to identification of inequity in the distribution of IC technologies and skills and its source across geographically determined groups of people in Poland. In the case of poor infrastructure or peoples’ skills, geographical inequity may increase, but it may also decrease if digital technologies were “effectively widely deployed to compensate for health workforce and health system deficiencies” [30]. In Poland, the infrastructure of healthcare had the tendency to be regionally diversified [60]. The World Health Organization underlines that “no one should be left behind–children or adults, rural or urban with digital solutions to improve their health and well-being” [35,49]. In the Polish health care system, the National Health Program (NHP) has also been formulated with a main goal being the elimination of geographical and social inequalities in health and limitation of social inequalities in health [61].

## 3. Materials and Methods

Data was derived from the Statistics Poland database for the period from 2017 to 2020 [62]. The range of research covers the whole of Poland. Administratively, Poland is divided into voivodeships, then counties and municipalities. According to Nomenclature of Territorial Units for Statistics (NUTS), Poland is divided into macro-regions (NUTS 1), regions (NUTS 2) and subregions (NUTS 3). All voivodeships (16) were regions according to NUTS, but in 2018 one voivodeship was divided into two regions (NUTS 2). As a result, a new seventh macro-region (NUTS 1) was introduced.

Thus, the unit of this research is region (voivodeship), which was determined by the availability of data. Finally, it allowed analyzing of the level of inequity in Poland in each of the macro-regions.

The range of variables was defined based on the literature [10,50] and then determined by availability of the Statistics Poland database. Finally, the following digital health determinants were used in this empirical research: household with and without Internet access, users and non-users of Internet and digital skills of people as follows: people with no digital skills then with, low, basic and higher than basic. All analyzed users and households are people in the age range from 16 to 74. In addition, variables which apply to characteristics of e-health usage (by people in the age from 16 to 74 and in the last 3 months) were employed, such as: searching for information about your own health or that of your relatives; arranging a medical visit via the website or application; access to medical documentation and using the services available through the website or app instead of visiting a doctor or hospital.

In order to analyze inequity, the Theil index was engaged. It was developed by Theil in 1967 and became the first important indicator to measure regional differences in economic development levels. The Theil index is a special form of the generalized entropy index system and nowadays, it is often used to measure spatial inequality [63]. The Theil index can take values from 0 to ∞, while the Theil index of 0 represents perfect equity while the value close to 1 presents the Pareto distribution.

Theil index is also one of the commonly used measures of inequity in the healthcare sector [64,65,66,67,68]. Other methods used are either Gini index [69,70,71,72,73] or concentration index [74,75,76]. Compared to other methods, the Theil index makes it possible to identify the main sources of the total differences by measuring the contribution of both intra- and inter-regional differences to the total inequity [77]. By using the Theil index, decomposition of the national overall inequality into between-macro-regions and within-macro-regions (between voivodeships/regions) differences was possible.

Formula (1) is used to calculate the total Theil index:
Theil index = ∑^*n*^_*i*=1_
*P_i_*ln(*P_i_*/*Y_i_*) (1)where:

*P_i_*—proportion of population aged 16–74 (household with people aged 16–74) in one voivodeship accounting for the total population aged 16–74 (households with people aged 16–74); and

*Y_i_*—proportion of population aged 16–74 (household with people aged 16–74) with the particular digital feature in one voivodeship accounting for the total population aged 16–74 (household with people aged 16–74) with the particular digital feature;

As Poland—the analyzed area—can be divided into smaller parts, such as seven macro-regions: central, northwestern, southwestern, northern, southern, eastern and Masovian voivodeship, thus the Theil index can be decomposed into the T_inter−class_ and T_intra−class_. Additionally, the contribution rates between and within groups can be calculated. If the proportion of T_intra−class_ is higher then it means that the inequity in digital variable distribution results more from the within-macro-regional difference and vice versa. Additionally, this is the same in the case of a higher proportion of T_inter−class_, which indicates that the inequity in digital variable distribution results more from the difference between each macro-region and vice versa.
Theil index = *T_intra−class_* and *T_inter−class_*(2)
*T_intra−class_* = ∑*^k^*_*g*=1_
*P_g_T_g_*
(3)
*T_inter−class_* = ∑^*k*^_*g*=1_
*P_g_*ln(*P_g_*/*Y_g_*)(4)where:

*T_intra_*_−*class*_—equity level of the particular digital variable distribution within the macro-regions;

*T_inter_*_−*class*_—equity level of the particular digital variable distribution between the macro-regions;

*T_g_*—Theil index of the particular digital variable distribution in macro-region;

*P_g_*—proportion of population in one macro-region accounting for the total population; and

*Y_g_*—proportion of digital variable in one macro-region accounting for the total digital variable

## 4. Results

The results of the Theil (T) index, shown in Table 1, indicate that households without Internet access and non-users of the Internet are not perfectly equally distributed in Poland. There are some macro-regions with more concentrated levels of limited access to the Internet. In addition, the value of Theil index increased from 2017 to 2020. However, households with access to the Internet and users of the Internet are almost equally distributed and there is also an improvement in 2020 as values are getting closer to 0.

Regarding the digital skills of the Polish population, the results show a relatively slight level of inequity as values vary in the range from 0.0035 to 0.0354 in the case of people with low, basic skills and higher than basic digital skills. However, there is higher deviation of Theil index values in the case of people with no digital skills as they are in the range from 0.0955 to 0.1470. Generally, it can be concluded that there are disparities in the distribution of people in Poland with different digital skills. However, the level of slight inequity decreased from 2017 to 2020 apart from people with low digital skills. These disparities in the distribution of digital resources and skills are also reflected in the unequal usage of eHealth.

A relatively lower equity can be observed in the access to medical documentation, followed by arranging a medical visit via the website, while people who search for information about their own health or their relatives’ health are relatively more equally distributed. However, the improvement can be observed in the case of arranging a medical visit via websites or applications. In 2017, the Theil index value deviated further from the value of 0 (it was 0.1780) than in 2020 (0.0865).

Application of the Theil index also allows showing the contribution of two components, i.e., the between macro-region component (T inter-class) and within-macro-regions component (T intra-class) to the national overall inequality/equity (T) (Table 2). In each year, within-macro-regions disparities are mainly responsible for all the slight unequal distributions of a particular variable. However, values vary from year to year in the case of digital skills. The between macro-region disparities are responsible for unequal distribution of people without digital skills from 2017 to 2019. The between-macro-regions inequity is the main factor for all slight inequities in the case of people with low digital skills in 2019 and people with higher than basic digital skills in 2019 and 2020. Therefore, it can be concluded that macro-regions are characterized by internal diversity, which is responsible for the existence of slight inequality at the level of the whole country.

In terms of macro-regional divisions (Table 3), the Theil index for the Masovian voivodeship shows relatively higher inside disparities in the distribution of analyzed variables than the remaining macro-regions in both years 2019 and 2020. There are some exceptions, because the northern macro-region was the region with the highest deviation from the equal distribution of people with no digital skills in 2020. Then, the southern and eastern macro-regions were characterized by the unequal distribution of people with basic digital skills in 2019 and 2020 appropriately. The southern macro-region was the one with the highest level of disparities in the case of people, who use the Internet to search for information about their own health or their relatives’ health.

It can be concluded, that the Masovian voivodeship is characterized by the highest degree of polarization. However, the growth change of Theil index values in many macro-regions for most of the analyzed variables is alarming and this phenomenon should be monitored. In the year of the pandemic the disparities increased slightly.

## 5. Discussion

This study has several major findings and merits. First, theoretical analysis showed that geographical equity in the distribution of information and communications technologies and skills matters from the health perspective.

Secondly, this empirical research identified some deviation from equity in distribution of the digital determinants of health and eHealth between geographically defined groups of the population. The values of Theil index took the range from 0.0003 to 0.2144 indicating some deviation from equity and thus slight inequities. However, in the case of health and health justice it is of high importance to eliminate any disparities as such deviation from the perfect equity can create the risk of discrimination in digital health. It is presented that in Poland, due to geographic status, people do not have a fair opportunity to achieve their full digital health potential. The macro-regions of Poland appeared to be slightly heterogeneous in terms of digital determinants of health distribution and eHealth. The main risk factors of health inequity are observed in the area of digital skills and usage of eHealth services, such as: arranging a medical visit via the website or application and then access to medical documentation, using the service available through the website or app instead of visiting a doctor or hospital. However, the usage of some eHealth services can also be limited by healthcare providers. As in the case of eHealth arrangement of a medical visit simply via the website or application where healthcare providers may not provide such opportunity.

Thirdly, these findings showed that the equity in distribution of digital determinants across the country is decomposable, and the disparities within-macro-regions represent an important part of the slight national inequalities, while, the between-macro-regions are only responsible for some national inequity in the case of the higher than basic digital skills of people in 2020. The results showed that there is no disproportion between western and eastern Poland, which was found in previous studies [78]. The reason for the dispersion within macro-regions may be the sparsity of population as was also found in the research by Bem et al. [79], Ucieklak-Jeż and Bem [80] and Rój [81] in the case of Poland (although this is also a problem found in other countries; see Wu, Song and Yu [82]). Most Poles with digital skills and information-communications technologies probably live in the large cities of the well-developed provinces. Thus, the results would be consistent with the current tendency of metropolization [83,84]. This problem, however, requires further analysis.

Fourthly, it was observed that the most diversified macro-region is the one with Warsaw, the capital city of Poland (Masovian), followed by the north-western and northern and south-western macro-regions. This could be the result of different rates of development leading to the growth of large centers and an increase of poorer surrounding areas, where dynamic economic growth is not taking place [85]. Thus, taking into account the complexity of digital variables and their interrelation with some other social determinants, the planning of a digital health strategy requires a comprehensive systematic approach.

Fifthly, due to the smaller significance of differences between individual macro-regions, the occurring disparities cannot therefore be explained through the prism of the historical factor as, in literature, this historical factor is often shown as the element responsible for differences in “local ties” and the economic attitudes of individual regions. In turn, they are the reasons why some regions differ in terms of social and economic resources and development [86,87,88,89]. However, this research has not supported this theory as the analysis of this factor in the context of the results obtained shows that the slight inequalities in the distribution of digital determinants cannot be explained through the prism of the existing differences in the social capital of individual regions and being caused by the historical factor.

However, these disparities can also be consequences of a still centralized model of health care management, therefore, all areas whose people cannot effectively take care of their health on their own. In such a situation, it is necessary, depending on the needs, to allocate appropriate intervention and financial resources to these areas.

As one of the two sources of health inequalities results from negligence or the health system policy, countries are therefore trying to reduce healthcare inequities and move toward solving the problem of achieving health justice by making carefully-planned policies and investment [2]. In light of the results obtained, Poland should also verify its health policy. These results showed a strong need to provide institutional support for some regions of Poland. National eHealth as well as digital health strategies should be more consolidated and supported by the appropriate action plans as well as resources. eHealth decision makers should consider the perspective of all parties, such as patients, health professionals, health-care providers, as well as the whole industry.

Some actions should be taken by the state and its institutions, such as the Ministry of Health and the National Health Funds (public payer of healthcare services). The National Health Funds should consider additional financing of some types of eHealth services to encourage providers to deliver them to patients. Thus, it is clear that government should support the development of e-health programs via health legislation with special attention to reimbursement of eHealth services legislation.

The results, which showed inequities, also confirm the need to facilitate the capacity for practitioners to use digital technologies for the purpose of delivering healthcare benefits effectively. Thus, some strategy should focus on the providers of health care for the purpose of encouraging them to make usage of the digital tools available to all patients, as well as but also to encourage patients to use digital health tools as part of standard care or to track digital health access and usage across sociodemographic variables. It would be important to focus on patient training in developing digital skills and the deployment of new technologies.

Therefore, the Ministry of Health should analyze the differences in the macro-regional variability of the behaviors of healthcare providers and their distribution as well as implement systematic solutions strengthening the training of human resources in health care, various professions and specialties (who can provide some support or encourage utilization of some forms of eHealth).

Another recommendation for health policy makers is the development and financing of training programs for health professionals and medicine students. These are important conditions for spreading the use of eHealth services in the country. An important condition for increasing the availability of e-services is the development of digital literacy in Poland of both patients and providers.

Therefore, it is necessary to accelerate work on the “Digital Competence Development Program until 2030”, on which work has been ongoing for two years. The aim of the program is to constantly raise the level of digital competences by providing everyone in Poland with the possibility of their development according to their needs. In addition, this program should precisely assign specific financial instruments to specific tasks and indicate a strict schedule for their implementation, as well as ensure its coordination with other plans of “The Polish Deal”.

The action plan of the Ministry of Health, for the present year, is too general in relation to telemedicine and eHealth. Such a plan should also be detailed and be clarified in terms of the rate and spatial scope of its implementation. It does not include the aspect of geographical equity.

In Poland, a new plan named “The Polish Deal” was introduced, which aims to reduce social inequities and create better living conditions for all citizens after the COVID period. According to it, the government wants to enable every household in Poland to have access to broadband Internet by 2024. This would mean that over a million more households are to gain broadband Internet access. However, the problem is that the government has not yet presented specific actions or strategies to achieve this goal. While such details are necessary, otherwise it will not be possible to achieve another goal, which is the development of the Patient Service Center (patient.gov.pl). This is the next stage of digitization of the health system, which will lead to a reduction in bureaucracy, and will make it possible to sign up to doctors, including specialists, via the Internet or a hotline. A necessary condition for the effective implementation of this goal is to ensure access to the Internet for everyone. Therefore, some corrections of both health policy and digitalization policy are required, any plans must be specified in detail and also coordinated. In addition, they must focus on improvement of both digital infrastructure and digital literacy of patients and providers. Such coordination actions will allow increasing of the efficiency of healthcare resources and decrease in its waste.

Consequently, this research confirmed that the geographical factor should be included in the process of regional health planning and digital technology planning as it is a key element of equity in the allocation of digital resources. This will allow one to ensure the universal accessibility of healthcare and thus sustainable development. This research also indicated the strong need of extending the range of collected data.

Future research may focus on an analysis of the relation between the health of the population and the distribution of digital determinants. Some further studies can also be conducted on the identification and investigation of these factors, which could explain both the level and the trends in the inequality of distribution of digital variables inside macro-regions. Moreover, as the research presented in this article focuses on Poland, which resulted from the purpose of the project—of which this article is part—therefore it will be of great importance to follow up on studies of other countries.

Based on the research it can be concluded that development of the digital area in health care must be planned and implemented precisely to protect people from unintended health inequity. The needs and capacity of the digital infrastructure and workforce should be thoroughly recognized and the strategy on digital health should then be reconstructed. This research underlines the importance of proper health equity data collection; otherwise, it would not be possible to monitor the health equity outcomes.

This study also faced some limitations. First, it was not possible to conduct county-level analysis because the available data was only disaggregated down to the voivodeship level. Secondly, it was not possible to perform an analysis separately for urban and rural areas of Poland because of the lack of data. It would allow for controlling their impact on the examined digital variables. Thirdly, the database allowed us to analyze the digital parameters only from the potential patients’ perspectives and not from the providers’ perspectives, while the World Health Organization provided a three-tier approach to digital health and its determinants, which means three perspectives—population, practitioner, and policymakers. Therefore, the scope of the collected data could be extended-by the perspective of practitioners. Furthermore, this study was limited to Poland as it results from the specific objectives of the project, under which the research is carried out.

## 6. Conclusions

This paper focuses on the equity distribution of information and communication technologies and skills. Any slight inequities in the distribution of digital health determinants are a serious threat to healthcare and sustainability. The results confirmed the existence of slight inequity and variability in the macro-regions of Poland. Geographical inequities should then be considered in the process of health and digital policy formulation. Thus, this research could be a base for healthcare decision makers in the process of correction of the unequal distribution of factors, such as digital resources, people skills and eHealth usage.

This study analyzed a certain range of digital determinants, which was limited by the availability of data. When the data is available, it would then be interesting to include—in the research—such variables as technology, infrastructure and financing, to recognize the overall status of digital resources and services in Poland and to extend this research to other countries.

## Figures and Tables

**Table 1 ijerph-19-02340-t001:** Theil index for the years from 2017 to 2020.

Variable	2017	2018	2019	2020
Households * with Internet access	0.0012	0.0010	0.0292	0.0003
Households * without Internet access	0.0236	0.0295	0.0259	0.0289
Users (aged 16–74) of Internet	0.0011	0.0009	0.0010	0.0007
Non-users (aged 16–74) of Internet	0.0203	0.0184	0.0308	0.0306
People (aged 16–74) with no digital skills	0.1308	0.0955	0.1470	0.1044
People (aged 16–74) with low digital skills	0.0092	0.0107	0.0079	0.0176
People (aged 16–74) with basic digital skills	0.0070	0.0035	0.0116	0.0041
People (aged 16–74) with higher than basic digital skills	0.0354	0.0346	0.0329	0.0312
eHealth **—searching for information about your own health or that of your relatives	0.0071	0.0100	0.0088	0.0064
eHealth **—arranging a medical visit via the website or application	0.1780	0.2144	0.1743	0.0865
eHealth **/***—access to medical documentation	-	-	-	0.1131
eHealth **/***—using the services available through the website or app instead of visiting a doctor or hospital	-	-	-	0.0567

Source: own calculation. * Households with people aged 16–74. ** People aged from 16 to 74, who used e-Health services in the last 3 months. *** Data started to be collected from 2020.

**Table 2 ijerph-19-02340-t002:** Theil index decomposition in the years from 2017–2020.

Variable	2017		2018		2019		2020	
	Inter	Intra	Inter	Intra	Inter	Intra	Inter	Intra
Households * with Internet access	0.0005	0.0007	0.0004	0.0006	0.0001	0.0291	0.0001	0.0002
Households * without Internet access	0.0091	0.0145	0.0101	0.0194	0.0042	0.0217	0.0099	0.0190
Users (aged 16–74) of Internet	0.0001	0.0010	0.0002	0.0007	0.0003	0.0007	0.0003	0.0003
Non-users (aged 16–74) of Internet	0.0025	0.0178	0.0031	0.0154	0.0086	0.0221	0.0140	0.0166
People (aged 16–74) with no digital skills	0.0849	0.0459	0.0502	0.0453	0.0813	0.0658	0.0451	0.0593
People (aged 16–74) with low digital skills	0.0029	0.0064	0.0043	0.0064	0.0059	0.0020	0.0087	0.0089
People (aged 16–74) with basic digital skills	0.0020	0.0050	0.0012	0.0023	0.0039	0.0077	0.0014	0.0027
People (aged 16–74) with higher than basic digital skills	0.0128	0.0226	0.0131	0.0214	0.0211	0.0118	0.0172	0.0140
eHealth **—searching for information about your own health or that of your relatives	0.0011	0.0060	0.0019	0.0062	0.0018	0.0070	0.0032	0.0033
eHealth **—arranging a medical visit via the website or application	0.0434	0.1346	0.0649	0.1495	0.0573	0.1170	0.0197	0.0668
eHealth **/***—access to medical documentation	-	-	-	-	-	-	0.0257	0.0873
eHealth **/***—using the services available through the website or app instead of visiting a doctor or hospital	-	-	-	-	-	-	0.0177	0.0390

Source: own calculation. * Households with people aged 16–74. ** People aged from 16 to 74, who used e-Health services in the last 3 months. *** Data started to be collected from 2020.

**Table 3 ijerph-19-02340-t003:** Theil index of macro-regions in Poland in the years from 2019 to 2020.

Variable/Macro-Region	Central		Northern		Northwestern		Southwestern		Southern		Masovian Voivodeship		Eastern	
	2019	2020	2019	2020	2019	2020	2019	2020	2019	2020	2019	2020	2019	2020
Households * with Internet access	0.0202	0.0001	0.0071	0.0000	0.0348	0.0003	0.0002	0.0000	0.0062	0.0000	0.0472	0.0006	0.0296	0.0003
Households * without Internet access	0.0022	0.0074	0.0011	0.0005	0.0051	0.0288	0.0000	0.0009	0.0011	0.0003	0.0072	0.0815	0.0048	0.0179
Users (aged 16–74) of Internet	0.0003	0.0000	0.0004	0.0000	0.0001	0.0005	0.0001	0.0003	0.0004	0.0002	0.0025	0.0010	0.0008	0.0000
Non-users (aged 16–74) of Internet	0.0094	0.0001	0.0134	0.0011	0.0041	0.0181	0.0045	0.0191	0.0136	0.0101	0.0877	0.0750	0.0126	0.0005
People (aged 16–74) with no digital skills	0.0037	0.0107	0.1106	0.1836	0.0699	0.0594	0.0003	0.0202	0.1359	0.0097	0.1626	0.1029	0.0072	0.0095
People (aged 16–74) with low digital skills	0.0014	0.0051	0.0055	0.0039	0.0010	0.0014	0.0000	0.0060	0.0013	0.0026	0.0219	0.0636	0.0011	0.0024
People (aged 16–74) with basic digital skills	0.0039	0.0019	0.0069	0.0011	0.0016	0.0012	0.0007	0.0029	0.0144	0.0003	0.0000	0.0002	0.0020	0.0060
People (aged 16–74) with higher than basic digital skills	0.0001	0.0056	0.0073	0.0074	0.0018	0.0079	0.0011	0.0014	0.0016	0.0016	0.0197	0.0323	0.0049	0.0060
eHealth **—searching for information about your own health or that of your relatives	0.0055	0.0000	0.0081	0.0007	0.0001	0.0047	0.0002	0.0006	0.0203	0.0169	0.0058	0.0067	0.0034	0.0027
eHealth **—arranging a medical visit via the website or application	0.0071	0.0011	0.0757	0.0479	0.0203	0.0092	0.0653	0.0389	0.0103	0.0104	0.3158	0.2611	0.0788	0.0618
eHealth **/***—access to medical documentation	-	0.0210	-	0.1014	-	0.0301	-	0.1480	-	0.0002	-	0.2615	-	0.0611
eHealth **/***—using the services available through the website or app instead of visiting a doctor or hospital	-	0.0259	-	0.0801	-	0.0031	-	0.0057	-	0.0273	-	0.1188	-	0.0079

Source: own calculation. * Households with people aged 16–74. ** People aged from 16 to 74, who used e-Health services in the last 3 months. *** Data started to be collected from 2020.

## Data Availability

The data presented in this study are available on request from the author.

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
