# Peer review of "Inequity in the Access to eHealth and Its Decomposition Case of Poland"

_ijerph, 2022, doi:10.3390/ijerph19042340_

Round 1

Reviewer 1 Report

Dear author

The topic use and the of Thiel Model to estimate inequalities in the provision of health facilities along with the ICT technologies is an appropriate topic in the present times. I would like the author to broaden the data base, if possible in some future such study. The present data evidence is restricted to Statistics Poland, so there will be some concerns about the general applicability of the study beyond polish borders. It may be desirable that the author mentions this in the limitations of the study and explain why the date was such restricted. I can understand one reason, i.e., the study is funded by a certain funder, so the funder may have preference in such matters.

Other than that, a moderate recheck and improvement in language usage and correctness in recommended.

I believe the consideration to, and incorporating above given suggestions will enhance the academic and practical value of the paper greatly.

Reviewer

Author Response

Dear Reviewer,

Thank you for your comments. I appreciate them so much. I included your comments in the article.

Point 1

I would like the author to broaden the data base, if possible in some future such study. The present data evidence is restricted to Statistics Poland, so there will be some concerns about the general applicability of the study beyond polish borders. It may be desirable that the author mentions this in the limitations of the study and explain why the date was such restricted. I can understand one reason, i.e., the study is funded by a certain funder, so the funder may have preference in such matters.

Response 1

I added information that the research was limited to Poland as this research is a part of the project which goal was limited to Poland (in the discussion section as limitation of study). I also added information about the direction of further research, which could also include other countries (in the discussion and conclusion part).

However, there are some studies in this area, which apply also only to one country and I added this info to the introduction section.

Point 2

Other than that, a moderate recheck and improvement in language usage and correctness in recommended.

Response 2

The text has now been corrected by a native speaker.

Reviewer 2 Report

In the introduction it is necessary to briefly present the gap in the literature that you want to fill in this paper and briefly present the research methods, objectives and results. You have only outlined the objectives.

You should be explaining what each section is all about in the paper.

Translate in english „powiats and gminas” or explain somehow this terms, if you can't translate it.

In conclusion, please present some solutions for policy makers based on the results and what would be the economic implications of these decisions.
It is clear that there are disparities in Poland, as everywhere else in the world, this is not new, but what are the solutions?
Is it enough to develop the infrastructure or is it also necessary to have some policies to increase literacy in Poland?

You have briefly presented these issues in the "Discussions" section, but it would be necessary to offer some concrete solutions based on the results of your research and especially to underline who should implement these solutions. 
Is it enough just to develop general health policies or is it also necessary to improve digitalization policies? 

Author Response

Dear Reviewer,

Thank you for your comments. I appreciate them so much. I included your comments in the article.

Point 1

In the introduction it is necessary to briefly present the gap in the literature that you want to fill in this paper and briefly present the research methods, objectives and results. You have only outlined the objectives. You should be explaining what each section is all about in the paper.

Response 1

In the introduction, I have briefly presented the gap in the literature and I have also outlined the research method and results. Then, I explained the content of each section in the paper.

Thus, four new paragraphs have appeared in the Introduction” section.

Point 2

Translate in english „powiats and gminas” or explain somehow this terms, if you can't translate it.

Response 2

I have translated them in english as „counties and municipalities”.

Point 3

In conclusion, please present some solutions for policy makers based on the results and what would be the economic implications of these decisions. It is clear that there are disparities in Poland, as everywhere else in the world, this is not new, but what are the solutions?

Is it enough to develop the infrastructure or is it also necessary to have some policies to increase literacy in Poland?

You have briefly presented these issues in the "Discussions" section, but it would be necessary to offer some concrete solutions based on the results of your research and especially to underline who should implement these solutions. Is it enough just to develop general health policies or is it also necessary to improve digitalization policies?

Response 3

In the discussion section, I have presented the proposition of concrete solutions in the area of health policy and digitalization policy and I also indicated who should introduce these solutions. I underlined also that the improvement of both digital infrastructure and digital literacy of patients and providers are needed.

Thus, five new paragraphs have appeared in the “Discussion” section.

Reviewer 3 Report

The article analyses disparities in distribution of information and communication technologies and skills with different population groups and aims at identifying source of the inequity. It discusses chances and pitfalls of eHealth. The discussion about eHeath is comprehensively covered and important quantitative indicators were identified.

I do not recommend the article to be published in this form. There are minor issues with language and discussion of methodology. I further doubt that the data supports the findings, however, as I am not an expert on the methodology of the Thiel index I recommend prove from the part of an expert.

The article analyses disparities in distribution of information and communication technologies and skills with different population groups and aims at identifying source of the inequity. It discusses chances and pitfalls of eHealth. The discussion about eHeath is comprehensively covered and important quantitative indicators were identified.

I do not recommend the article to be published in this form. There are minor issues with language and discussion of methodlogy. I further doubt that the data supports the findings, however, as I am not an expert on the methodology of the Thiel index I recommend prove from the part of an expert.

Introduction:

The article could be better introduced. The first sentence “Everyone has the right to enjoy the highest attainable standard of health” should indicate the origins of this right, where does it come from - human rights and WHO.

Data and data interpretation:

I am not so much into the specifics of the Theil Index, but it seems as it is used appropriately. the It is true, that the Theil index can take values from “from 0 to ∞” (p 4). However, I think that it is important that 1 represent the Pareto-distribution. The values all vary within a very small range, with only few exceptions under 0.1. Therefore, I am not sure whether it is possible to identify any temporal tendencies with the differences being so small.

Furthermore, it is too shortly discussed how the values of the eHealth variables are connected to the inequity variables, as a causal influence from one to the other is not proven by the methods. The method just shows a correlation. I am thus not convinced that the study “showed that geographical equity in the distribution of information and communications technologies and skills matters from the health perspective” (p 8).

According to my understanding, the data does not show any serious inequities. Only “E-health** – arranging a medical visit via the website or application” shows values around 0.2  with 0.1780 in 2017,  0.2144 in 2018, 0.1743 in 2019 which dropped to 0.0865 in 2020. It would be helpful to have a discussion whether the variation in values is significant. Is there a test for significance? Probably other inequality indicators, such as Gini, could be used to evaluate the significance.

As I am not a very expert in inequality metrics, I suggest the interpretation being checked by another reviewer.

Formalities and Language:

In table 1 values are sometimes written as 0.0002 and sometimes with comma such as 0,0002

Table 3 how it is formatted is unreadable.

To me phrases such as “it is a high importance” (p 3) seem odd, as I would say it is of high importance” however I cannot say whether this is wrong or just not within my English skills.

Same with “it is also underline …” (p 3) and shouldn’t it be “people’s skills” instead of “people skills” (p 3)?

Definitely “ofently” (p 4) should be “often” and “witihn-macro-regions” should read “within-macro-regions”. I suggest the text being gone through by a native speaker.

Author Response

Dear Reviewer,

Thank you for your comments. I appreciate them so much. I included your comments in the article.

Introduction:

Point 1

The article could be better introduced. The first sentence “Everyone has the right to enjoy the highest attainable standard of health” should indicate the origins of this right, where does it come from - human rights and WHO.

Response 1

I corrected the first sentence and I indicated the origin of this sentence – Constitution of the World Health Organization.

Data and data interpretation:

Point 2

I am not so much into the specifics of the Theil Index, but it seems as it is used appropriately. the It is true, that the Theil index can take values from “from 0 to ∞” (p 4). However, I think that it is important that 1 represent the Pareto-distribution. The values all vary within a very small range, with only few exceptions under 0.1. Therefore, I am not sure whether it is possible to identify any temporal tendencies with the differences being so small.

Response 2

I added information about meaning of “1” as the value of Theil index (i.e. The Pareto distribution) in the ”Materials and Methods” section.

Indeed, the Theil index values vary within very small range, what reflects the particular state and the lack of equity and of course not tendency. I have changed some elements - in the “Results” and “Discussion” sections - in purpose to avoid so strong statement as „tendency”, which maybe introduced some confusion.

Point 3

Furthermore, it is too shortly discussed how the values of the eHealth variables are connected to the inequity variables, as a causal influence from one to the other is not proven by the methods. The method just shows a correlation. I am thus not convinced that the study “showed that geographical equity in the distribution of information and communications technologies and skills matters from the health perspective” (p 8).

Response 3

I have corrected the above sentence as it can be misleading. By this sentence, I wanted to express that in theories I found the arguments for the importance of geographical equity from the perspective e-Health i.e. distribution of information and communications technologies. This sentence does not apply to empirical results. It is a “finding” of analysis of theory (conclusion of „theoretical background' section of this article).

Therefore, I have changed this sentence appropriately.

Point 4

According to my understanding, the data does not show any serious inequities. Only “E-health** – arranging a medical visit via the website or application” shows values around 0.2 with 0.1780 in 2017, 0.2144 in 2018, 0.1743 in 2019 which dropped to 0.0865 in 2020. It would be helpful to have a discussion whether the variation in values is significant. Is there a test for significance? Probably other inequality indicators, such as Gini, could be used to evaluate the significance.

As I am not a very expert in inequality metrics, I suggest the interpretation being checked by another reviewer.

Response 4

Indeed, the results do not show any serious (strong) inequities. The Theil index values vary within very small range but from the other hand, it does not mean that there is perfect equity. Results confirm the existence of disparities and in the context of health (health justice) any deviation from the equity (0 value) matters.

However, I have calculated the Gini for my data and Gini coefficients took the values from 0.01 to 0.3, which indeed confirm that there are not alarming inequities (based on the numbers). It is not so bad but is not very good or perfect.

The application of Theil index allowed me to make the decomposition of equity/inequity, while the Gini Coefficient allows only to measure the overall equity / inequity. The goal of the article / research was to find also the source of overall inequity therefore, the Gini method does not allow to realize this purpose.

Therefore, in purpose to avoid the above confusion, I have changed some strong statement in the article such as “relatively high inequity” etc for softer and more adequate statements.

Also, according to you suggestion, I have added some discussion whether this variation in values is significant taking into account the perspective of numbers and these numbers in the context of health and health justice.

Formalities and Language:

Point 5

In table 1 values are sometimes written as 0.0002 and sometimes with comma such as 0,0002

Response 5

It has been corrected.

Point 6

Table 3 how it is formatted is unreadable.

Response 6

I made sure that format of Table 3 is readable. In case I am sending also pdf version of article.

Point 7

To me phrases such as “it is a high importance” (p 3) seem odd, as I would say it is of high importance” however I cannot say whether this is wrong or just not within my English skills.

Same with “it is also underline …” (p 3) and shouldn’t it be “people’s skills” instead of “people skills” (p 3)?

Definitely “ofently” (p 4) should be “often” and “witihn-macro-regions” should read “within-macro-regions”. I suggest the text being gone through by a native speaker.

Response 7

The text has been checked by a native speaker and all above problems and others have been corrected.
